# The Effect of Salt Reduction on Sensory, Physicochemical, and Microbial Quality in Selected Meat Products

**DOI:** 10.3390/foods14183150

**Published:** 2025-09-09

**Authors:** Miroslav Jůzl, Josef Kameník, Alena Saláková, Blanka Macharáčková, František Ježek, Markéta Janík Piechowiczová, Jan Slováček

**Affiliations:** 1Department of Food Technology, Faculty of AgriSciences, Mendel University in Brno, Zemědělská 1665/1, 613 00 Brno, Czech Republic; miroslav.juzl@mendelu.cz (M.J.); 24160@node.mendelu.cz (A.S.); xpiechow@node.mendelu.cz (M.J.P.); 2Department of Animal Origin Food and Gastronomic Sciences, University of Veterinary Sciences Brno, Palackého třída 1946/1, 612 42 Brno, Czech Republic; kamenikj@vfu.cz (J.K.); macharackovab@vfu.cz (B.M.); jezekf@vfu.cz (F.J.); 3The Czech Trade Inspection Authority, tř. Kpt. Jaroše 1924/5, 602 00 Brno, Czech Republic

**Keywords:** sodium, reformulation, colour, sensory analyses, frankfurter, sausage, špekáčky

## Abstract

Efforts to reduce sodium chloride levels in meat products within the European Union are closely aligned with the World Health Organization’s recommendation to limit daily sodium intake to a maximum of 5 g per person. The nutritional significance of excessive salt intake is well-established, and the role of sodium chloride in meat products is not only crucial for sensory quality but also for hygienic safety. The aim of this study was to evaluate the impact of varying salt concentrations (1.4%, 1.6%, 1.8%, and 2.0%) on selected quality parameters of two common processed meat products: grilling sausages (Špekáčky) and frankfurters (Wiener Würstel). Sensory analysis was conducted with a relatively high number of trained evaluators (n = 73 for Wiener Würstel; n = 63 for Špekáčky). The results indicated that moderate salt reduction had no statistically significant effect (*p* > 0.05) on either the sensory attributes or the hygienic quality of the products. The method of salt content determination was shown to be critical for accurate assessment. These findings suggest that both types of reformulated products with reduced sodium content can be effectively used in production without compromising quality. Therefore, such formulations may serve as viable strategies for the food industry to contribute to public health objectives by reducing dietary sodium intake.

## 1. Introduction

In the meat industry, salt is not used solely as a flavour enhancer but also for its wide range of functional properties. Sodium chloride (NaCl) solubilizes myofibrillar proteins, thereby activating them and enhancing their water-binding capacity, texture-forming ability, and emulsifying capacity, which together contribute to improved product stability [1,2]. For effective protein activation, a minimum concentration of 12 g NaCl per kg of meat is typically required [3]. At the same time, a salt level of approximately 1.0% is generally considered the threshold for acceptable flavour in processed meat products [4], as insufficient salting often results in a bland, unappetizing taste [5].

Salt also plays a critical role in food preservation, especially in meat products, where it inhibits microbial growth by lowering water activity and, in combination with other preservation factors, creates an unfavourable environment for spoilage bacteria and pathogens [6,7,8]. For example, it has been reported that the salt content in cooked sausages and cooked hams can be reduced by about 0.5% without significantly affecting shelf life [9].

Nevertheless, the excessive intake of dietary sodium has become a major global public health concern. Scientific evidence has increasingly linked high sodium consumption to hypertension and cardiovascular diseases [10], and average global salt consumption still exceeds the WHO’s recommended maximum of 5 g per person per day [11]. Europeans continue to exceed the recommended salt intake by more than twofold [12].

Consequently, the reformulation of processed foods, including meat products, is being promoted as an effective tool to improve public health outcomes [13]. Reformulation strategies targeting sodium reduction in meat products include partial substitution of sodium chloride with potassium chloride, magnesium sulphate, calcium-based salts, seaweed, fruit juice concentrates, or bacterial cultures—some of which may also allow for simultaneous nitrite reduction. However, these strategies often lead to variable outcomes in terms of product quality, safety, and consumer acceptance [14,15,16].

Although the reduction of sodium in meat products has been extensively investigated, most studies have primarily focused on whole-muscle products or alternative reformulation strategies. Currently, according to some published authors [17], approximately 20% of dietary sodium intake originates from meat products. However, evidence regarding traditional Central European cooked sausages, such as frankfurters (Wiener Würstel) and grilling sausages (Špekáčky), remains limited, despite the fact that these belong to the most frequently consumed processed meats in this region. In Central Europe, the average consumption of processed meat products remains high, ranging between 25 and 35 kg per capita annually, with sausages and frankfurters representing a dominant category. Reducing the salt content in these traditional products is challenging; nevertheless, they are often perceived positively from both a social and nutritional perspective [18]. In addition, consumer attitudes play a role in shaping dietary patterns. Respondents with stronger environmental concerns were more likely to adopt vegetarian or vegan diets, whereas more traditional and conservative consumers tended to prefer meat-based or flexitarian eating habits [19]. Consequently, these products constitute a major dietary source of sodium for the population and represent a suitable model system for the evaluation of salt reduction strategies.

The aim of this study was to evaluate the impact of varying sodium chloride concentrations (1.4%, 1.6%, 1.8%, and 2.0%) on the physicochemical, sensory, and microbiological properties of cooked meat products filled into natural casings. This work provides evidence on the feasibility of moderate salt reduction in two traditional and culturally significant product categories, with relevance for both industry practice and public health. Importantly, such findings may also influence the willingness of food companies to implement reformulation strategies. However, any attempt to reduce sodium levels must carefully balance technological feasibility and consumer acceptance, since inappropriate changes may negatively affect product quality parameters and ultimately hinder consumer approval.

## 2. Materials and Methods

### 2.1. Production of Selected Meat Products

The experiment was conducted at the Meat Pilot Plant Laboratory, Department of Food Technology, Mendel University in Brno. Boneless beef and pork were purchased from Jatka Ivančice, s.r.o. (CZ 242 ES, Ivančice, Czech Republic), transported in a refrigerated truck under chilled conditions, with the temperature at delivery not exceeding +4 °C. Meat was stored at 2.5 °C and processed within 48 h post-slaughter. Standard additives (spices, curing mixture with nitrite) were obtained from MasoProfit (Brno, Czech Republic).

Two types of cooked meat products were prepared in two independent production batches: grilling sausages (Špekáčky) and frankfurters (Wiener Würstel). Both are traditional Central European sausages, differing in formulation, structure, and culinary use.

Grilling sausages (Špekáčky) are traditional Czech and Slovak sausages characterized by visible chunks of pork back fat, typically filled into natural beef casings with a diameter of 40/43 mm and grilled over an open fire. This product is legally defined in the Czech Republic (Decree No. 69/2016 Coll.) [20] and recognized at the EU level under the Traditional Specialities Guaranteed (TSG) scheme (EU 2016) [21]. The standard formulation (per 100 kg of product, brutto) consisted of 38.5 kg beef (≤30% fat), 17.5 kg pork (≤50% fat), 27.0 kg pork back fat (bacon), and 23.0 kg ice water, 2.5 kg potato starch, and a spice mixture. The spice mixture consisted of 0.22 kg ground sweet paprika (100 ASTA), 0.16 kg ground black pepper, 0.09 kg garlic concentrate, 0.03 kg ground nutmeg, 0.30 kg polyphosphates (E450, E451), and 0.05 kg ascorbic acid (E300). Four different formulations were prepared with curing salt concentrations of 1.4%, 1.6%, 1.8%, and 2.0%, with the 2.0% group serving as the control. The 2.0% concentration was selected because it represents a commonly applied salt level in processed meat products, particularly in frankfurters and grilling sausages [1,22]. This decision was also supported by data from a comparative study of Czech and German meat products, which reported similar sodium levels in these categories [12]. Using this reference point enabled us to align the experiment with standard industrial practice and to systematically evaluate the effects of stepwise salt reduction on sensory, physicochemical, and microbiological properties. The meat batter was prepared using a vacuum bowl cutter (K 64 Ultra VA, Seydelmann, Stuttgart, Germany) and followed the standard technological procedure: in the first phase, lean meat was chopped with curing salt, ice, and the spice mixture. Subsequently, high-fat components, including frozen pork back fat, were incorporated [23]. The final batter for Špekáčky was mixed at lower speeds to preserve a coarse structure with visible fat chunks. After completing the comminution and mixing process, potato starch was added to the final meat batter to achieve the desired consistency and binding properties. The batters were filled into natural beef casings (40/43 mm), with sausage portions separated by string tying using a vacuum filler HTS 95 (HTS Fleischereimaschinen, Thalgau, Austria).

Frankfurters (Wiener Würstel) are finely comminuted cooked meat products traditionally served warm, often as part of a hot dog. Their formulation (per 100 kg) included 20.0 kg beef (≤30% fat), 13.3 kg pork (≤20% fat), 40.0 kg pork (≤50% fat), and 23.5 kg ice water. The combination of leaner and fattier pork reflects standard industrial practice, as it allows achieving the desired texture, juiciness, and sensory attributes of the product, while beef improves firmness and structural stability [22,24]. A commercial spice mixture (MASOPROFIT, EAN: 8592355611106) was added at 9.5 g/kg, consisting of ground spices (sweet paprika, ginger, coriander, nutmeg), polyphosphates (E451), monosodium glutamate (E621), ascorbic acid (E300), and paprika extract (E120). The meat batter was produced in two independent batches using the same vacuum bowl cutter (K 64 Ultra VA, Seydelmann, Germany), initially processing the lean meat with salt, ice, and spices. Subsequently, the high-fat pork components were incorporated. Unlike Špekáčky, the frankfurter batter was processed into a fine, homogeneous emulsion, characteristic of emulsified meat product types [25]. The filling was performed using a vacuum filler (HTS 95, HTS Fleischereimaschinen, Austria) into sheep casings with a diameter of 20/22 mm. As with the grilling sausages, four salt concentrations (1.4%, 1.6%, 1.8%, and 2.0%) were applied, with the 2.0% formulation used as the control.

All products were heat-treated in a smoking chamber (Bastramat B 850 FR, Bastra GmbH, Arnsberg, Germany) until reaching a core temperature of 70 °C (maintained for 10 min), which corresponds to the minimum recommended value to ensure microbiological safety, particularly regarding the inactivation of *Salmonella* spp. and *Listeria monocytogenes* [5], monitored by a temperature probe. After cooking, the sausages were rapidly cooled to +5 °C, vacuum-packed (Henkelman Boxer 35 P, Henkelman Vacuum Systems, The Netherlands), and stored at +2 °C until analyses were conducted (within three weeks of production).

### 2.2. Sensory Evaluation

Sensory evaluation was conducted at Mendel University in Brno and the University of Veterinary Sciences Brno, using trained panellists according to ISO 8586 (2023) [26]. Evaluations were carried out at the midpoint of the declared shelf life (three weeks after production) in a sensory room under standardised ISO 6658 (2017) conditions [27].

Unstructured 100 mm graphical line scales were used, anchored by verbal descriptors at both ends (0 = extremely poor, 100 = excellent). The use of unstructured 100 mm line scales in sensory evaluation is a well-established method in food sensory science [23,25]. The following attributes were evaluated: appearance on cut, colour, cohesiveness, odour, consistency, texture, saltiness, intensity of saltiness, flavour, and overall impression. Samples were anonymised and identified by random three-digit codes.

Triangle tests were also performed to assess the panel’s ability to distinguish saltiness. Two identical samples and one differing in salt content were presented, and panellists were asked to identify the odd sample. The combinations tested were 1.4% vs. 2.0%, and subsequently 1.6% vs. 1.8%. Finally, panellists were instructed to rank coded samples in order of increasing salt content.

A total of 73 panellists evaluated the grilling sausages (Špekáčky), and 63 panellists evaluated the frankfurters (Wiener Würstel), aged 19–67 years.

Prior to evaluation and colour measurement, samples were heat-treated to simulate typical consumption. Grilling sausages were grilled in a combi oven (Rational SCC WE 61, Rational AG, Germany) at 140 °C with 0% humidity (fully open vent) for 10 min. Frankfurters were steam-heated at 80 °C and 100% humidity (closed vent) for 10 min.

### 2.3. Instrumental Colour Measurement

*L**, *a**, and *b** values were measured with a CM-3500d spectrophotometer (Konica Minolta, Japan; illuminant D65, 6500 K) on both the surface and cross-section in SCE mode (Specular Component Excluded) using an 8 mm slot. Heat-treated (HT) samples were cooled to 20–25 °C prior to measurement, under the same conditions as described for the sensory evaluation. Untreated (UT) samples were likewise tempered to 20–25 °C to ensure consistency in measurement conditions. Each sample was measured five times. The overall colour change was expressed as ΔE*_ab_ [28].

The total colour difference (ΔE*_ab_), where ∆*L**, ∆*a** and ∆*b** were values related to the control group with 2% salt, was calculated using the formula:$$\Delta E{*}_{\mathrm{ab}}\;=\;\surd (\Delta L^{*2}+\Delta a^{*2}+\Delta b^{*2})$$

### 2.4. Chemical Analysis

**Dry matter** was determined by drying 10 g of sample to constant weight at 103 ± 2 °C [29].

**Fat content** was measured by Soxhlet extraction (Soxtec 2055, FOSS, Höganäs, Sweden) using petroleum ether as the solvent.

**Crude protein** content was determined using the Kjeldahl method (ISO 937:2002) on a Kjeltec 2300 analyser (FOSS, Höganäs, Sweden), using a nitrogen-to-protein conversion factor of 6.25 [30].

Determination of the **salt content over chlorides** was performed by the titration method after sample leaching with water and determining all chlorides, which were then recalculated to sodium chloride [31].

Samples were taken for **determination of the sodium concentration** to verify the proportion of salt in the samples of meat products. The sodium content was determined by atomic absorption spectrometry. To digest the meat product, 6 mL of concentrated nitric acid (65% *v*/*v*) and 2 mL of hydrogen peroxide (30% *v*/*v*) were added to 0.25 g of the sample and mineralised using an Ethos SEL Microwave Labstation (Milestone, Milan, Italy) at 200 °C for 30 min. The sodium content was then measured using air-acetylene flame atomisation in a contrAA 700 atomic absorption spectrometer (Analytik Jena, Jena, Germany). All samples were measured in triplicate and the values obtained were processed by Aspect CS software, version 2.1, resulting in one final value for each sample (batch of product). The Na-based salt content (in %) was calculated by applying a conversion coefficient of 2.5 in accordance with Regulation (EU) No. 1169/2011 [32].

### 2.5. Microbiological Analysis

Microbiological analyses were conducted on samples examined two days post-packaging and after 14 days of refrigerated storage at 2 °C, targeting total viable counts and lactic acid bacteria. For each salt concentration level, five individual product units were analysed. Results are expressed as the mean of these five replicates.

Basic microbiological sample preparation was carried out in accordance with ISO standards [33,34]. A 25 g portion of the sample was aseptically transferred into a sterile stomacher bag and homogenized with 225 mL of sterile buffered peptone water (BPW; Oxoid, UK).

Samples were investigated for the presence of the total number of microorganisms (total viable count, TVC) and lactic acid bacteria (LAB).

When detecting the total number of bacteria, mesophilic aerobic and facultative aerobic microorganisms were enumerated using a non-selective, nutrient-rich medium incubated aerobically at 30 °C for 72 h [35]. The pour plate method was employed to determine the total microbial count, expressed as colony-forming units (CFUs) per gram of sample. As the culture medium, glucose, tryptone, and yeast extract agar (GTK; Oxoid, UK) were used.

Determination of lactic acid bacteria was conducted on a nutrient-rich medium (MRS agar; Oxoid, UK) under anaerobic conditions, with incubation at 30 °C for 72 h [36]. Colonies displaying distinct morphological characteristics were selected from each sample and subjected to catalase and oxidase tests (JK Trading, Prague, Czech Republic). The results are expressed as colony-forming units (CFUs) per gram of sample.

### 2.6. Statistical Analysis

All experiments were conducted on two independent production batches (n = 2). The number of replicates for each analysis is specified in the respective methodological subsections. Microbiological data (TVC, LAB) were log-transformed prior to analysis. Data normality and homoscedasticity were checked (Shapiro–Wilk; Levene). Between-group differences were tested by two-way ANOVA (product type × salt level); where applicable, batch was treated as a random effect. Post hoc comparisons used Tukey’s HSD (α = 0.05). Triangle tests were evaluated by the exact binomial test, and ranking data by the Friedman test with Nemenyi post hoc. Microbiological results are presented descriptively, as no statistically significant differences were observed between treatments. Statistical analyses were performed using Statistica 14 (TIBCO, Santa Clara, CA, USA).

## 3. Results and Discussion

### 3.1. Sensory Analysis

The results of the sensory evaluation of meat products with varying salt content are presented in Table 1. A statistically significant difference (*p* < 0.05) was observed only for the descriptor saltiness intensity, and this was true for both types of meat products. In the case of grilling sausages, the variant containing 1.6% salt received the lowest score for saltiness. For frankfurters, a significant difference (*p* < 0.05) was found between the 1.4% salt variant and the 2.0% and 1.6% salt variants, with the 1.4% sample rated as less salty. These findings indicate that only the most pronounced differences in salt concentration (e.g., 1.4% vs. 2.0%) were consistently perceived by the sensory panel. Smaller stepwise reductions (0.2 percentage points) appear to fall close to the discrimination threshold for this type of product, which explains the absence of significant differences in most descriptors. This is further supported by the triangle and ranking tests, which demonstrated that panelists were able to detect salt differences at the extremes, but struggled when variations were small. Such patterns are consistent with previous studies reporting that moderate sodium reductions may be sensorially compensated by the product matrix, fat-to-protein ratio, and seasoning profile [1,2,3,4].

Fat is known to negatively influence the release of sodium during consumption, thereby reducing perceived saltiness [37]. These effects, sometimes described as masking, can be explained by the complex interaction between fat, protein, and salt in the product matrix. While fat itself tends to attenuate salt perception by limiting sodium release during mastication, products with a higher fat-to-protein ratio may still exhibit a relatively stronger salty taste compared to leaner formulations, as protein has been shown to suppress saltiness perception. In our study, this attenuating effect of fat may have contributed to the limited perceptual differences among the intermediate salt concentrations. The interaction between fat and salt should therefore be considered when interpreting the sensory results, as it may explain the absence of significant differences in several descriptors despite stepwise salt reductions. It should also be noted that individual sensory descriptors may have different levels of importance for different consumer groups. As the population ages, particularly in Europe, it may become necessary to adjust weighting factors or modify the perception of sensory quality for specific descriptors in certain foods. Bañón et al. [38] identified the main causes of sensory deterioration as including rancid, bitter, and mouldy flavours, increased hardness, and loss of juiciness in meat products.

No statistically significant differences (*p* > 0.05) were observed in the remaining descriptors. However, from a general standpoint, cross-sectional appearance was rated best for the 1.4% salt variant of grilling sausages, while the 2.0% salt variant was rated the lowest. For frankfurters, differences in the appearance of the cross-section were minimal. In terms of the colour descriptor, the 1.4% salt variant of grilling sausages received the lowest score, while no marked differences were recorded between groups in frankfurters.

Regarding flavour, the 1.6% and 1.8% salt variants of grilling sausages were rated the highest, whereas the 2.0% variant received the lowest rating. For frankfurters, the 1.8% and 2.0% variants were rated most favourably. Overall sensory acceptability in grilling sausages followed the same pattern as flavour ratings, while no substantial differences were observed between the frankfurter groups.

Salt enhances the flavour intensity of meat products; however, saltiness perception is also modulated by other factors, such as the salt-to-fat ratio. Products with higher fat content generally exhibit a stronger salty taste, whereas increased protein content (i.e., more lean meat) tends to suppress saltiness perception [22].

In the triangle test evaluating grilling sausages with the greatest salt content difference (1.4% vs. 2.0%), 64% of panellists correctly identified the odd sample, while 36% responded incorrectly. In a second triangle test comparing smaller salt differences (1.6% vs. 1.8%), 52% of responses were correct and 48% incorrect.

For frankfurters, the triangle test involving the largest salt difference (1.4% vs. 2.0%) yielded 62% correct and 38% incorrect responses. In the second test (1.6% vs. 1.8%), 44% of answers were correct, and 56% incorrect.

Another sensory test involved ranking the samples by saltiness intensity, from the least salty to the saltiest. This test was evaluated based on the number of correctly ordered samples per assessor. In 26% of the cases for grilling sausages and 19% for frankfurters, none of the samples were placed in the correct order. Conversely, 16% of panellists correctly ranked all grilling sausage samples, and 11% did so for frankfurters. Two samples were placed in the correct order by 33% of panellists for both products, while only one sample was placed correctly by 25% (grilling sausages) and 37% (frankfurters) of assessors.

Zandstra et al. [13] reported that consumer acceptance of low-sodium foods can increase, provided that the overall sodium intake from the rest of the diet remains unchanged. Several studies have shown that salt content in specific products can be gradually reduced over time in small increments, and that a total reduction of 20–30% may remain within acceptable sensory thresholds, though this is often product-specific. Our results support the feasibility of positive consumer acceptance of meat products with reduced salt content.

Nevertheless, practical limitations need to be considered when reducing salt in meat products. Reformulation may increase production costs due to the use of compensatory ingredients (e.g., potassium salts, flavour enhancers), and subtle changes in sensory profile can still affect consumer acceptance, particularly in traditional products. Moreover, alternatives such as KCl may impart off-flavours, limiting their use. Previous studies have highlighted that gradual stepwise reductions are more feasible and better accepted by consumers compared to abrupt changes [22,39].

### 3.2. Instrumental Colour Measurement

The results of the instrumental colour measurements of the meat products, as affected by salt content, are presented in Table 2 and Table 3. Statistically significant differences (*p* < 0.05) in surface lightness (*L**) were observed among frankfurters before heat treatment, with the control group containing 2.0% NaCl differing from all reduced-salt formulations. In grilling sausages before heat treatment, significant differences (*p* < 0.05) were found for the *b** parameter (yellowness) between the control and all other samples with lower salt concentrations. According to López-López et al. [40], a decrease in *L** and *b** values caused by formulation changes was negatively perceived by consumers, indicating the sensitivity of colour perception to compositional modifications.

Following heat treatment, no statistically significant differences (*p* > 0.05) were observed in any of the CIELab colour parameters on the surface of the tested products. However, the overall colour difference (ΔE*_ab_) relative to the 2.0% NaCl control tended to increase as salt content decreased. Despite this trend, the observed changes generally remained within the range of slight to moderate visual differences. Appearance and colour are among the most influential attributes during initial consumer assessment and are often compared to an internally anchored reference standard, consciously or unconsciously employed during purchase decisions [25]. Nevertheless, consumer judgement is multifactorial, and absolute colour measurements obtained through instrumental methods may not fully capture perceptual responses. Therefore, sensory evaluation remains essential when assessing the consumer relevance of such differences [9,13].

The colour of the cut surface (Table 3) was particularly influenced in frankfurters, where statistically significant differences (*p* < 0.05) in *L** values were recorded between the control (2.0% NaCl) and salt-reduced variants, both before and after heat treatment. Reduced-salt frankfurters consistently exhibited a lighter appearance on the cross-section.

The ΔE*_ab_ values, when compared to the control, mostly indicated moderate and clearly perceptible colour deviations in both product types. A pronounced difference in ΔE*ab was detected in heated treated frankfurters containing 1.8% salt. This variation did not follow the expected linear trend associated with stepwise salt reduction and was not confirmed by sensory analysis (*p* > 0.05). Similar discrepancies were described by Jůzl et al. [25], who highlighted the critical role of recipe quality, including the type and proportion of meat, in shaping the sensory perception of frankfurters.

In summary, the colour changes induced by salt reduction were generally minor and did not compromise product appearance to a critical degree. These alterations likely reflect a complex interplay of other formulation components, including phosphate levels and spice mixture, which are influenced by the type of meat used, manufacturer-specific recipe traditions, and quality management practices. Finally, consumer perceptions of colorants and flavour enhancers, as potential markers of “unnaturalness”, may influence acceptability and purchase intent, even when such additives serve a technological function [23].

### 3.3. Chemical Analysis

The fat and protein content in the tested products primarily depended on the composition of the raw materials. The levels determined in frankfurters were in accordance with previously published values [41,42]. Table 4 summarizes the results of dry matter, fat, protein, and sodium content, recalculated as sodium chloride (NaCl) content, as determined by both chloride ion titration and atomic absorption spectrometry (AAS), in relation to the salt levels used in product formulations. Notably, the relationship between fat and sodium content may be more complex than merely the practice of salting higher-fat products to a greater extent than leaner formulations. Vulič et al. [43]. suggested that products with a higher fat-to-muscle ratio are often associated with elevated sodium levels, indicating a potential compositional linkage beyond processing practices. Moreover, as demonstrated by Kameník et al. [12], the intrinsic sodium fraction present in muscle tissue can contribute to differences in salt content depending on the analytical method applied. This highlights the need to interpret sodium values not only in terms of added salt, but also with consideration of the raw material composition and the methodological approach used for determination.

The NaCl content in grilling sausage samples determined by AAS ranged from 1.77% to 1.86%, whereas titration-based chloride analysis yielded values from 1.48% to 2.07%. In frankfurters, the AAS method indicated a range from 1.58% to 2.08%, while chloride titration ranged from 1.49% to 2.08%. The titration method confirmed a statistically significant (*p* < 0.05) salt concentration gradient in both product types, with only minor deviations (3–7%) from the target values. In contrast, AAS results for sausages did not reflect a significant difference (*p* > 0.05) among groups, with deviations from target salt levels reaching up to 26%. For frankfurters, AAS-based salt values also varied in accordance with the formulation, albeit with a higher deviation (up to 13%) compared to titration. These discrepancies underline the critical importance of selecting a suitable analytical method for salt determination, especially when assessing compliance with sodium-reduction targets.

The notably higher dry matter content observed in grilling sausages with the lowest salt concentration (1.4%) may be attributed to a reduced water-binding capacity of the meat matrix, which has previously been associated with lower salt formulations [42,44].

It is essential to note that sodium in comminuted meat products originates not only from sodium chloride but also from other sodium-containing additives such as sodium triphosphate, sodium metabisulfite, sodium nitrate, and sodium bicarbonate. Thus, the correlation between sodium (Na) and sodium chloride (NaCl) content is product-specific [15]. Intrinsically, muscle tissue contains less than 100 mg Na/100 g [45], whereas the primary source of sodium in processed meats remains NaCl, comprising 39.3% sodium by mass. Additional sodium contributions may arise from additives like monosodium glutamate, sodium citrate, and sodium lactate. Although the sodium load from these compounds is generally lower than from NaCl, their cumulative impact must not be overlooked. From a public health perspective, the overarching aim of meat producers is to reduce sodium content in their products, thus supporting healthier dietary patterns [1].

The compositional parameters of the grilling sausages, particularly fat and protein content, were consistent with data reported by Conroy et al. [46]. Among the samples, the dry matter content was significantly higher (*p* < 0.05) in the formulation with 1.4% salt, reaching 45.26 ± 0.90%, while no significant differences were observed among frankfurters. Fat and protein content did not vary significantly (*p* > 0.05) in either product type.

Capuano et al. [47], analysing sodium in 1016 food samples, reported the highest values in bacon (1370 mg Na/100 g) and frankfurters (1330 mg Na/100 g). Similarly, Webster et al. [48], in a study involving 7221 commercial food items grouped into 90 subcategories, identified processed sausages and spreads as containing the highest sodium concentrations (1283 mg/100 g), followed by other processed meat products (846 mg/100 g).

In terms of product functionality, the water-holding capacity of the meat matrix is reflected in fat retention. Products with higher fat content may also display increased susceptibility to lipid oxidation, particularly when the salt content is elevated. Common salt levels in processed meats range from 1 to 2%. However, excessive salt concentrations may exert pro-oxidative effects on lipids. These effects are mediated via multiple pathways, including osmotic disruption of cellular membranes, release of iron ions from heme and non-heme proteins, and inhibition of endogenous antioxidant enzymes [49]. Although salt-reduced products exhibit a lower risk of lipid oxidation, this can be further mitigated by incorporating natural antioxidants. Such strategies not only enhance oxidative stability but are also generally well accepted by health-conscious consumers [50].

### 3.4. Microbiological Assessment

At the beginning of the study, the total viable count (TVC) of the examined samples was determined on the second day after packaging to assess the initial microbial contamination, reaching 3.63 log CFU/g (grilling sausages = 3.23 log CFU/g; frankfurters = 3.84 log CFU/g) (Table 5). The difference in TVC depending on the salt content was not statistically significant and ranged between 3.04 and 3.49 log CFU/g in grilling sausages and between 3.56 and 4.08 log CFU/g in frankfurters. Statistical testing confirmed the absence of significant treatment effects (*p* > 0.05); therefore, microbiological results are primarily presented descriptively in Table 5. After 14 days of refrigerated storage, a decrease in bacterial counts was observed, reaching approximately 3.11 log CFU/g (grilling sausages = 2.73 log CFU/g; frankfurters = 3.32 log CFU/g) (Table 5). Again, differences in TVC related to salt content were not statistically significant (grilling sausages = 2.49–3.04 log CFU/g; frankfurters = 3.20–3.46 log CFU/g). Based on the study by Kreyenschmidt et al. [51] the progression of microbial growth in vacuum-packed meat products stored under refrigeration followed the following pattern: ham > turkey breast fillet > smoked pork loin > pariza > mortadella > bacon and ham > frankfurters.

Although meat product manufacturing involves processing of raw materials naturally contaminated during slaughter and handling, a significant portion of contamination is indirect and occurs via the production environment. This is controlled by strict hygienic regulations, disinfection protocols, and HACCP (Hazard Analysis and Critical Control Points)-based procedures. In this context, the reduction of salt, which is a key preservative, may pose a considerable challenge in maintaining microbial stability. However, Talon et al. [52] reported that technological microbiota on surfaces and equipment residues are commonly composed of *Staphylococcus*, *Kocuria*, and LAB (lactic acid bacteria), with levels ranging from undetectable to high, exceeding 8.0 log CFU/100 cm^2^. According to the authors, LAB are generally less present on surfaces than microorganisms of the genera *Staphylococcus* and *Kocuria*.

Lactic acid bacteria (LAB), which in non-fermented products may represent a spoilage risk, were present in the range of 1.15 to 2.62 log CFU/g on the second day after packaging (grilling sausages = 2.15 log CFU/g; frankfurters = 2.04 log CFU/g) (Table 5). Despite conditions that would typically favour LAB proliferation, no significant increase in LAB counts was observed after 14 days of refrigerated storage. On the contrary, in grilling sausages with 1.4% and 1.6% salt, a decrease in LAB counts was recorded (1.4%: from 1.95 to 1.40 log CFU/g; 1.6%: from 2.62 to 0.60 log CFU/g) (Table 5). A reduction in LAB counts during cold storage was also observed in frankfurters at all salt levels. Although the samples were vacuum-packed prior to analysis, previous studies [53] have shown that hygienic safety depends on multiple factors and is not solely determined by packaging type. Nevertheless, vacuum packaging remains one of the most widely used methods in the meat industry due to its ability to limit microbial growth. Interestingly, a decrease in both TVC and LAB counts was observed after 14 days of chilled storage. Such reductions have also been reported in heat-treated, vacuum-packed meat products and may result from several factors. These include batch-to-batch variability, natural fluctuations of microbial populations, and the combined inhibitory effects of residual nitrite, reduced salt levels, and refrigeration. In addition, sublethally damaged cells from thermal processing may gradually lose viability during storage before subsequent regrowth occurs [22]. This initial decline does not compromise product safety and is consistent with known microbiological dynamics in processed meat products.

Although consumer preference, product type, and other factors must be considered, there is clear potential for salt reduction in meat products [54]. Such efforts are well justified. A 25% salt reduction, which aligns with WHO targets based on national consumption averages, is a reasonable threshold towards which reformulation studies should aim [7].

Multiple factors need to be considered to further improve the quality of salt-reduced meat products [17]. In recent years, several European countries have amended their legislation in favor of lowering sodium levels in processed foods, including meat products. Experts therefore highlight a combination of strategies—legislative measures, mandatory or voluntary sodium labeling, consumer education, and gradual reformulation—as essential for effectively reducing sodium intake from processed meat [55]. However, achieving this goal remains complex and requires coordinated action beyond the sole responsibility of either producers or consumers.

Although sodium intake in the Czech Republic exceeds WHO recommendations, the actual salt content of foods, including meat products, is comparable to that in other European countries. For example, Kameník et al. [12] reported no significant difference in sodium levels between Czech and German cooked sausages, despite the fact that overall sodium intake in Germany is more than twice as high [56]. This highlights the importance of considering broader dietary habits and lifestyle factors when interpreting salt intake data. Furthermore, Dušková et al. [57] demonstrated that the sodium content in cooked sausages can be reduced to approximately 1.7% without compromising shelf life or product safety, which is in line with other studies showing that combinations of additional technological interventions may have synergistic or compensatory effects.

## 4. Conclusions

This study evaluated the impact of stepwise sodium chloride reduction (2.0–1.4%) on the quality and safety of two traditional Central European meat products: frankfurters (Wiener Würstel) and grilling sausages (Špekáčky). The results showed that moderate reductions in salt had no major effect on product safety or overall sensory quality, with only saltiness intensity being significantly affected. Instrumental colour and proximate composition remained stable, and microbiological counts decreased or remained unchanged during refrigerated storage, confirming the hygienic stability of the products.

The novelty of this work lies in its focus on traditional Central European products, for which detailed reformulation data have been lacking. Our results demonstrate that sodium levels can be reduced without compromising consumer-relevant traits, and we recommend a concentration of 1.6–1.8% NaCl as optimal for both product types. This range provides a balance between sensory acceptability, technological feasibility, and public health objectives, and can serve as a practical guideline for meat processors aiming to meet consumer expectations for lower-salt products. Future research should also include instrumental texture analysis to provide a more comprehensive evaluation of quality changes associated with sodium reduction.

## Figures and Tables

**Table 1 foods-14-03150-t001:** Results of the sensory evaluation of grilling sausages (Špekáčky) and frankfurters (Wiener Würstel) with different salt concentrations (Mean ± S.E.).

Descriptor	Grilling Sausages (n = 73)				Frankfurters (n = 63)			
	1.4%	1.6%	1.8%	2.0%	1.4%	1.6%	1.8%	2.0%
**appearance on cut**	76 ± 20	72 ± 20	74 ± 20	70 ± 20	82 ± 16	83 ± 16	79 ± 20	81 ± 16
**colour**	75 ± 19	77 ± 19	79 ± 17	77 ± 17	81 ± 16	81 ± 17	80 ± 15	80 ± 16
**cohesiveness**	69 ± 20	67 ± 22	67 ± 19	61 ± 19	79 ± 17	80 ± 17	78 ± 18	78 ± 17
**odour**	74 ± 19	77 ± 19	81 ± 16	74 ± 20	80 ± 17	82 ± 15	81 ± 16	79 ± 17
**consistency**	79 ± 15	80 ± 16	78 ± 16	75 ± 18	82 ± 17	82 ± 16	78 ± 18	80 ± 17
**texture**	80 ± 17	82 ± 16	78 ± 16	80 ± 17	76 ± 20	73 ± 20	72 ± 21	75 ± 20
**saltiness**	72 ± 20	72 ± 19	73 ± 22	67 ± 21	70 ± 20	69 ± 20	68 ± 22	71 ± 21
**intensity of salt**	59 ± 18 ^a^	50 ± 17 ^b^	53 ± 15 ^ab^	55 ± 17 ^ab^	50 ± 19 ^b^	54 ± 18 ^ab^	56 ± 18 ^a^	63 ± 18 ^a^
**flavour**	72 ± 21	74 ± 20	75 ± 19	69 ± 22	74 ± 19	75 ± 19	78 ± 19	79 ± 17
**overall impression**	66 ± 22	71 ± 19	69 ± 19	63 ± 20	69 ± 19	69 ± 20	71 ± 19	71 ± 18

Different superscripts in the rows within one product type indicate statistically significant differences (*p* < 0.05).

**Table 2 foods-14-03150-t002:** Colour properties on surface of grilling sausages (Špekáčky) and frankfurters (Wiener Würstel) with different salt concentrations before and after heat treatment (Mean ± S.E.).

Product Type	Salt Content	*L**	*a**	*b**	dE*_ab_	*L**_HT	*a**_HT	*b**_HT	dE*_ab__HT
**Grilling** **sausages**	1.4%	44.88 ± 1.04	21.71 ± 0.48	29.85 ± 0.90 ^a^	2.56	45.57 ± 6.84	18.48 ± 3.23	17.52 ± 4.28	2.26
	1.6%	45.15 ± 1.34	21.13 ± 0.78	29.13 ± 0.78 ^a^	2.24	45.38 ± 6.12	20.36 ± 3.73	21.37 ± 5.53	6.43
	1.8%	45.48 ± 2.17	21.59 ± 0.91	30.08 ± 1.06 ^a^	3.28	47.17 ± 4.77	18.37 ± 3.07	16.60 ± 5.69	1.51
	2.0%	43.73 ± 0.64	21.88 ± 1.17	27.57 ± 1.08 ^b^	0.00	46.29 ± 7.46	18.14 ± 2.62	15.40 ± 4.35	0.00
**Frankfurters**	1.4%	50.98 ± 1.40 ^a^	20.06 ± 0.48 ^c^	26.47 ± 0.81	4.07	48.31 ± 0.59	22.56 ± 0.68	27.03 ± 0.79	1.56
	1.6%	51.81 ± 0.99 ^a^	21.72 ± 0.69 ^b^	26.62 ± 1.24	3.79	49.14 ± 1.39	22.00 ± 0.82	26.14 ± 0.70	0.71
	1.8%	51.64 ± 0.90 ^a^	21.41 ± 0.53 ^b^	27.10 ± 0.80	3.67	47.98 ± 2.13	21.99 ± 0.88	26.69 ± 1.06	1.13
	2.0%	48.35 ± 0.61 ^b^	22.98 ± 0.21 ^a^	27.50 ± 0.47	0.00	48.67 ± 1.36	21.51 ± 0.55	25.94 ± 0.97	0.00

Different superscripts in the columns within one product type indicate statistically significant differences (*p* < 0.05). Variant with 2.0% of salt is the control for dE*_ab_, HT—heat treatment.

**Table 3 foods-14-03150-t003:** Colour properties on cut of grilling sausages (Špekáčky) and frankfurters (Wiener Würstel) with different salt concentrations before and after heat treatment (Mean ± S.E.).

Product Type	Salt Content	*L**	*a**	*b**	dE*_ab_	*L**_HT	*a**_HT	*b**_HT	dE*_ab__HT
**Grilling** **sausages**	1.4%	63.32 ± 2.28 ^a^	12.92 ± 2.09 ^b^	12.03 ± 1.20	4.62	46.21 ± 6.85	19.04 ± 3.29	19.25 ± 5.13	4.04
	1.6%	60.35 ± 2.12 ^b^	13.58 ± 1.27 ^b^	12.35 ± 0.63	1.73	48.17 ± 6.17	17.65 ± 3.42	18.09 ± 4.08	4.46
	1.8%	60.74 ± 1.57 ^b^	13.57 ± 1.23 ^b^	12.57 ± 0.88	2.08	50.80 ± 5.96	16.26 ± 2.56	14.79 ± 3.59	6.74
	2.0%	59.07 ± 0.94 ^b^	14.73 ± 0.54 ^a^	12.13 ± 0.49	0.00	44.40 ± 6.96	18.13 ± 4.25	15.76 ± 4.68	0.00
**Frankfurters**	1.4%	59.67 ± 0.59 ^a^	15.48 ± 0.19 ^ab^	12.34 ± 0.29 ^a^	5.14	58.36 ± 0.58 ^a^	16.07 ± 0.19 ^a^	13.14 ± 0.24 ^a^	3.84
	1.6%	58.24 ± 0.83 ^a^	15.80 ± 0.28 ^ab^	12.75 ± 0.22 ^a^	3.82	58.16 ± 0.76 ^a^	15.23 ± 0.33 ^b^	12.49 ± 0.46 ^b^	3.30
	1.8%	58.49 ± 0.63 ^a^	15.88 ± 0.20 ^a^	12.78 ± 0.16 ^a^	4.08	58.14 ± 0.48 ^a^	15.16 ± 0.40 ^bc^	12.70 ± 0.40 ^a^	3.31
	2.0%	54.57 ± 1.69 ^b^	15.40 ± 0.41 ^b^	11.77 ± 0.44 ^b^	0.00	54.94 ± 0.57 ^b^	14.76 ± 0.19 ^c^	11.98 ± 0.35 ^b^	0.00

Different superscripts in the columns within one product type indicate statistically significant differences (*p* < 0.05). Variant with 2.0% of salt is the control for dE*_ab_, HT—heat treatment.

**Table 4 foods-14-03150-t004:** Chemical properties of grilling sausages (Špekáčky) and frankfurters (Wiener Würstel) with different salt concentrations (Mean ± S.E.).

Product Type	Salt Content	Dry Matter %	Fat %	Protein %	NaCl—Based Cl^−^ %	NaCl—Based Na^+^ %
**Grilling** **sausages**	1.4%	45.26 ± 0.97 ^a^	29.15 ± 1.24	11.21 ± 1.03	1.48 ± 0.08 ^d^	1.77 ± 0.16
	1.6%	43.46 ± 0.32 ^b^	27.23 ± 2.61	11.12 ± 0.54	1.68 ± 0.08 ^c^	1.80 ± 0.12
	1.8%	43.70 ± 0.81 ^b^	26.52 ± 4.67	11.43 ± 0.68	1.85 ± 0.07 ^b^	1.80 ± 0.11
	2.0%	43.09 ± 0.79 ^b^	28.04 ± 3.27	11.25 ± 0.72	2.07 ± 0.04 ^a^	1.86 ± 0.18
**Frankfurters**	1.4%	35.11 ± 0.76	14.64 ± 1.55	14.87 ± 1.08	1.49 ± 0.06 ^d^	1.58 ± 0.09 ^d^
	1.6%	34.19 ± 0.57	13.37 ± 1.66	15.70 ± 0.50	1.65 ± 0.03 ^c^	1.92 ± 0.10 ^c^
	1.8%	33.11 ± 1.73	12.47 ± 2.44	15.55 ± 1.12	1.85 ± 0.04 ^b^	2.05 ± 0.10 ^b^
	2.0%	33.02 ± 2.88	12.86 ± 3.01	15.62 ± 0.46	2.08 ± 0.04 ^a^	2.08 ± 0.02 ^a^

Different superscripts in the columns within one product type indicate statistically significant differences (*p* < 0.05).

**Table 5 foods-14-03150-t005:** Total Viable Counts (TVCs) and lactic acid bacteria (LAB) in grilling sausages (Špekáčky) and frankfurters (Wiener Würstel) with different salt concentrations after production and at the end of storage of 14 days (Mean).

Product Type	Salt Content	TVC (log CFU·g^−1^)	TVC_14 (log CFU·g^−1^)	LAB (log CFU·g^−1^)	LAB_14 (log CFU·g^−1^)
**Grilling** **sausages**	1.4%	3.08	3.04	1.95	1.40
	1.6%	3.49	2.49	2.62	0.60
	1.8%	3.11	2.51	2.43	2.08
	2.0%	3.04	2.61	1.15	1.40
**Frankfurters**	1.4%	3.73	3.34	2.48	1.04
	1.6%	3.82	3.20	2.04	N.D.
	1.8%	4.08	3.46	N.D.	N.D.
	2.0%	3.56	3.26	1.11	1.08

N.D.—not detected.

## Data Availability

The original contributions presented in the study are included in the article, further inquiries can be directed to the corresponding author.

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
