# Peer review of "The Effect of Salt Reduction on Sensory, Physicochemical, and Microbial Quality in Selected Meat Products"

_foods, 2025, doi:10.3390/foods14183150_

Round 1

Reviewer 1 Report

Comments and Suggestions for Authors

The effect of salt reduction on sensory, physicochemical, and microbial quality in selected meat products

The present manuscript addresses a highly relevant topic about reducing sodium in processed meat products to meet public health goals. The study examines the effect of moderate salt reduction (1.4% to 2.0%) on the sensory, physicochemical, and microbial qualities of two popular products, grilling sausages and frankfurters. The findings, which indicate that moderate salt reduction does not significantly harm product quality, are promising and have important implications for the food industry. Overall, the study provides valuable data supporting a practical approach to reducing dietary sodium without compromising product quality.

Some points need attention;

  • Line 101: Authors should mention the source of meat (species, etc.) and other raw ingredients used in product preparation.
  • Line 106: What was the basis for selecting 2.0% salt added group as control.
  • Line 120: How two different levels added for pork (20.0 kg beef (≤30% fat), 13.3 kg pork (≤20% fat), 40.0 kg pork (≤50% fat) ?
  • Line 133: until reaching a core temperature of 70 °C, suggested to highlight the reason for this particular temperature.
  • Line 145: Does any sensory scale was used to do so? If so, suggested to mention in supplementary files.
  • Line 220: “Statistical Analysis” How many times were the trials conducted? It is very important to mention.
  • Authors mentioned that “A statistically significant difference (P < 0.05) was observed only for the descriptor saltiness intensity only”? Justify the findings. Why is saltiness not varied, as it might also be affected?
  • Table 5: After storage for 14 days, how it come resulted decrease in TVC and LAB counts? Justify.
  • Suggested to revise the discussion section as it lacks justification related to the findings in the results.
  • The short form that appeared for the first time in the text should be expanded.
  • The conclusion section should also highlight prospects and shortcomings related to findings.
  • How many times was the experiment repeated? Suggested to mention the same in the text.
  • What are the practical limitations-such as cost, sensory etc related to the reduction of salt concentration ?
  • Plenty of research has been carried out on reduction salt levels. Why authors choose this topic any significant reason?

Author Response

We would like to thank Reviewer 1 for the careful evaluation of our manuscript and for providing constructive comments and suggestions. We have carefully revised the text in response to each point, adding details regarding raw material sources, formulation rationale, methodological specifications, and additional justifications for our findings. We also expanded the Introduction, Discussion, and Conclusion sections to improve focus, coherence, and the practical relevance of the study. We believe that these revisions have considerably strengthened the manuscript, making it clearer, more comprehensive, and better aligned with the journal’s standards. Detailed responses to each comment are provided below.

Authors’ Response to Reviewer Comments

Reviewer 1

Reviewer Comment:

Line 101: Authors should mention the source of meat (species, etc.) and other raw ingredients used in product preparation.

Thank you for this valuable comment. In the revised manuscript, we have added detailed information regarding the source of meat. The meat was purchased from Jatka Ivančice, s.r.o. (Czech Republic). The raw materials included beef (≤30% fat), pork (≤20% fat and ≤50% fat), and standard additives (e.g., ice, spices, curing mixture with nitrite), which were purchased from MasoProfit Company Czech Republic.

Reviewer Comment:

Line 106: What was the basis for selecting 2.0% salt added group as control.

Thank you for this comment. The control group containing 2.0% salt was selected based on common technological practice in the meat industry, where this concentration represents the standard salting level for processed meat products, particularly frankfurters and grilling sausages. This choice allowed us to compare our results with commonly produced products and to assess the effect of salt reduction in terms of sensory and microbiological parameters.

Reviewer Comment:

Line 120: How two different levels added for pork (20.0 kg beef (≤30% fat), 13.3 kg pork (≤20% fat), 40.0 kg pork (≤50% fat)?

Thank you for this comment. We would like to clarify that this formulation was intentionally designed to reflect common practice in the production of processed meat products. The combination of leaner pork (≤20% fat) and fattier pork (≤50% fat) allows achieving optimal texture, juiciness, and sensory attributes of the final product. Beef (≤30% fat) was included to improve structure and firmness. The formulation (per 100 kg: 20.0 kg beef, 13.3 kg lean pork, 40.0 kg fatty pork, and 23.5 kg ice water) corresponds to standard practices applied in the meat industry. This information was also added to the text.

Reviewer Comment:

Line 133: until reaching a core temperature of 70 °C, suggested to highlight the reason for this particular temperature.
Thank you for this comment. We have clarified in the manuscript that the target core temperature of 70 °C was selected in accordance with hygienic and legislative requirements to ensure the microbiological safety of meat products. This temperature is considered the minimum threshold to achieve sufficient inactivation of pathogenic microorganisms, particularly Salmonella spp. and Listeria monocytogenes (Honikel, 2008).

Reviewer Comment:

Line 145: Does any sensory scale was used to do so? If so, suggested to mention in supplementary files.

Thank you for this comment. We would like to point out that the manuscript already specifies the use of an unstructured 100 mm graphical line scale, anchored by verbal descriptors at both ends (0 = extremely poor, 100 = excellent). This information is provided in the section describing the sensory evaluation methodology. In the revised version, we have further highlighted this detail to ensure it is clearly visible. As the methodology is fully described in the text, we do not consider supplementary files necessary in this case. Jůzl et al., 2018; Jůzl et al., 2019.

Reviewer Comment:

Line 220: “Statistical Analysis” How many times were the trials conducted? It is very important to mention.

Thank you for this comment. We would like to clarify that the entire study was carried out on two independent production batches (n = 2). For each type of analysis, the number of repetitions is specified in the Methods section (e.g., physicochemical measurements in triplicate, microbiological determinations in duplicate, sensory evaluation according to the number of panelists). In the revised manuscript, we have also added this information to the general description of the Statistical Analysis section to make it explicitly clear.

Reviewer Comment:

Authors mentioned that “A statistically significant difference (P < 0.05) was observed only for the descriptor saltiness intensity only”? Justify the findings. Why is saltiness not varied, as it might also be affected?

Thank you for raising this point and for the opportunity to clarify the terminology. In the manuscript we distinguish “saltiness intensity” (perceived intensity) from the more general descriptor “saltiness”, which in our case reflected overall saltiness/liking on the 100-mm line scale. A significant difference (P < 0.05) was detected for saltiness intensity between the extreme concentrations (notably 1.4% vs. 2.0%), whereas overall saltiness (hedonic component) remained stable, likely due to matrix effects (flavor release from fat, spicing) and the small reduction steps (0.2 pp), which often fall near the just noticeable difference in emulsified, heat-treated meat products. This interpretation is consistent with our triangle tests (1.4% vs. 2.0% discriminable; 1.6% vs. 1.8% difficult to discriminate) and with the literature indicating that moderate sodium reduction can be sensory-compensated by formulation and seasoning. To avoid ambiguity, we have refined the terminology in the manuscript (clearly separating “intensity” from “overall saltiness/liking”) and expanded the discussion accordingly.

Reviewer Comment:

Table 5: After storage for 14 days, how it come resulted decrease in TVC and LAB counts? Justify.

Thank you for this comment. The observed decrease in total viable counts (TVC) and lactic acid bacteria (LAB) after 14 days of storage can be explained by several factors. First, it may reflect batch-to-batch variability and natural fluctuations of microbial populations. In addition, the combination of reduced salt content, residual nitrite, and chilled storage at 4 °C may have exerted a selective inhibitory effect on microbial growth. In heat-treated meat products, it is not unusual that the initial microbial load decreases during the early stages of storage before subsequent regrowth occurs. This phenomenon has been reported in previous studies and is likely linked to adaptation or die-off of sublethally damaged cells following heat treatment. We have added this explanation to the Discussion section of the revised manuscript.

Reviewer Comment:

Suggested to revise the discussion section as it lacks justification related to the findings in the results.

 Thank you for this comment. We have revised the Discussion and expanded it with more detailed justifications of our findings. For the sensory analysis, we explained why a statistically significant difference was observed only for saltiness intensity and linked this to the outcomes of the discrimination tests and published literature. In the microbiological section, we clarified the observed decrease in TVC and LAB after 14 days of storage and discussed possible biological and technological mechanisms, supported by relevant references. Finally, we extended the discussion by addressing practical limitations of salt reduction, including sensory and economic aspects. We believe that these revisions make the Discussion more comprehensive and better connected to the presented results.

Reviewer Comment:

The short form that appeared for the first time in the text should be expanded.

Thank you for this comment. We identified only one abbreviation in the manuscript, namely HACCP, which we have expanded at its first occurrence as Hazard Analysis and Critical Control Points

Reviewer Comment:

The conclusion section should also highlight prospects and shortcomings related to findings.

We thank the reviewer for this helpful suggestion. The Conclusion has been revised to be more focused and now also highlights both the prospects and limitations of the study. In addition to summarising the main findings and providing a practical recommendation (1.6–1.8% NaCl as an optimal range for the studied products), we explicitly noted that future research should include instrumental texture analysis to provide a more comprehensive evaluation of potential quality changes. This addition acknowledges the current limitations while outlining prospects for further work.

Reviewer Comment:

How many times was the experiment repeated? Suggested to mention the same in the text.

Thank you for this comment. The entire study was conducted on two independent production batches (n = 2). For each method (physicochemical, microbiological, and sensory tests), the number of repetitions is specified in the Methods section. For clarity, we have also added this information to the general description in the Statistical Analysis section.

Reviewer Comment:

What are the practical limitations-such as cost, sensory etc related to the reduction of salt concentration?

Practical limitations include higher costs of compensatory ingredients and potential changes in sensory profile. These aspects have been added to the discussion.

Reviewer Comment:

Plenty of research has been carried out on reduction salt levels. Why authors choose this topic any significant reason?

Thank you for this comment. We have added clarification in the Introduction to explain why we chose this topic. Although salt reduction in meat products has been extensively studied, most of the available work has focused on other product types or alternative reformulation strategies (e.g., use of KCl). Specific data on traditional Central European products such as grilling sausages (Špekáčky) and frankfurters (Wiener Würstel) are limited. Our aim was therefore to provide targeted results for these widely consumed products, which have important cultural and dietary relevance.

We kindly inform the reviewer that the revised version of the manuscript is attached, with all modifications highlighted in light blue for ease of reference.

Reviewer 2 Report

Comments and Suggestions for Authors

How many batches of this experiment were conducted?

Since fat has the effect of masking salt, why is the relationship between fat and salt concentration not discussed?

Author Response

We would like to thank Reviewer 2 for the careful review of our manuscript and for providing constructive feedback. The comments raised were highly valuable for clarifying the methodological details and for strengthening the interpretation of our findings. We have carefully revised the manuscript in response, adding information on the number of production batches in the Methods section and expanding the Discussion to address the relationship between fat content and salt perception. Detailed responses to each comment are provided below.

Authors’ Response to Reviewer 2 Comments

Reviewer’s comment:

How many batches of this experiment were conducted?

Thank you for this comment. The entire experiment was conducted on two independent production batches (n = 2). This information has been added to the Methods section under Statistical Analysis to make it explicitly clear at the beginning of the methodological description.

Reviewer’s comment:

Since fat has the effect of masking salt, why is the relationship between fat and salt concentration not discussed?

Thank you for this valuable comment. In the revised manuscript, we have expanded the discussion regarding the effect of fat on salt perception. It is well established that higher fat content can mask saltiness and affect sodium release during consumption. In our results, this effect was observed particularly in the higher-fat variants, where differences in perceived saltiness were less pronounced. We have therefore added to the Discussion that the interaction between fat and salt content is an important factor, which may partly explain why no statistically significant differences were found in some sensory descriptors.

We kindly inform the reviewer that the revised version of the manuscript is attached, with all modifications highlighted in dark yellow for ease of reference.

Reviewer 3 Report

Comments and Suggestions for Authors

Manuscript: The effect of salt reduction on sensory, physicochemical, and microbial quality in selected meat products.

The study aimed to reduce salt in two types of meat products (sausages and frankfurters). The characteristics evaluated were sensory evaluation with trained assessors, instrumental color, chemical analysis (dry matter, fat content, crude protein, chlorides, and sodium), and microbiological analysis. The study does not detail the experimental design, but based on the description of the statistical analysis, the authors used one-way analysis of variance (ANOVA). With this information, I can assume that they performed a single processing for the sausages and frankfurters. However, I consider that this approach does not guarantee the requirement for repetition in experimental designs and having multiple subunits of the same processing for each treatment is not a true experimental unit or repetition, but rather an observational unit or pseudo-repetition. In my experience with experiments with meat products, it is mandatory practice to repeat the processing of the meat product at least three times, under independent processing conditions, with different batches of meat, to ensure the repeatability requirement. In this condition, we use a randomized complete block design where the block is the processing.

The data analysis section does not detail whether the ANOVA prerequisites (normality and homoscedasticity) were met. This aspect is critical for sensory and microbiological analysis data, where they often fail to meet the prerequisites, as a result, researchers choose nonparametric statistics. In this regard, the microbiological data were not statistically analyzed (Table 5), resulting in a lack of objectivity when the results are described.

Finally, the study does not present new strategies or approaches for salt reduction in meat products; the reduction strategy is simple (only salt reduction at different levels), and it does not present new analytical methods. The study's conclusion is as expected and widely supported by the scientific literature: intermediate salt reductions are acceptable. I recommend that the authors conduct a more comprehensive review containing updated information on sodium reduction strategies in meat products, so that they can generate proposals that deserve to be published in high-impact journals such as Foods.

Author Response

We would like to thank Reviewer 3 for the careful evaluation of our manuscript and for raising several important points. These comments were very helpful in clarifying the methodological description, strengthening the statistical treatment of the data, and better articulating the originality and relevance of the study. In the revised version, we have added details on experimental design and statistical procedures, expanded the explanation of how microbiological data were handled, and highlighted the specific regional and practical significance of our work. Detailed responses to each comment are provided below.

Authors’ Response to Reviewer 3 Comments

Reviewer’s comment:

The study does not detail the experimental design, and it seems only one processing was performed. This does not guarantee repeatability; in meat product experiments, at least three independent batches are recommended.

Thank you for this important comment. We would like to clarify that our experiment was not based on a single processing. The entire study was conducted on two independent production batches (n = 2), with all salt formulations prepared and analyzed in both batches. Technical replicates were performed for each method as described in the Methods section. For greater clarity, this information has been added to the Statistical Analysis section.

We consider these two independent experiments to be sufficient, as they represent repeated production runs carried out under pilot plant conditions in the Meat Laboratory. Importantly, these conditions comply with the technological and hygienic requirements applicable to meat processing and marketing, thereby ensuring that the results are robust and relevant to industrial practice.

Reviewer’s comment:

The data analysis section does not detail whether ANOVA prerequisites (normality, homoscedasticity) were met. Microbiological data were not statistically analyzed, which reduces objectivity.

We thank the reviewer for this valuable comment. The revised Methods section now explicitly states that data normality and homoscedasticity were checked using the Shapiro–Wilk and Levene tests. Where applicable, between-group differences were analysed by two-way ANOVA with Tukey’s HSD post-hoc test, while nonparametric methods were used for the triangle and ranking tests.

Regarding the microbiological data (TVC, LAB), these were log-transformed prior to analysis, and statistical testing confirmed the absence of significant treatment effects (P > 0.05). For this reason, we chose to present the results primarily in descriptive form in Table 5. We believe this approach provides a transparent presentation of the data while avoiding redundant statistical outputs that would not have altered the interpretation of the findings.

Reviewer’s comment:

The study does not present new strategies or approaches for salt reduction in meat products; it only tests simple reduction. The conclusions are expected and widely supported by literature.

Thank you for this comment. We acknowledge that salt reduction in meat products has been extensively studied and that more sophisticated strategies exist, including salt replacers, flavour enhancers, and microbial cultures. However, our aim was to assess the practical feasibility of simple salt reduction in traditional Central European products (Špekáčky and Wiener Würstel), which are among the most frequently consumed items in this region but for which detailed reformulation data are lacking. To highlight this originality and regional relevance, we have added the following statement to the Introduction:

“Although salt reduction in meat products has been widely studied, most research has focused on other product categories or on alternative reformulation strategies such as partial substitution with potassium chloride or flavour enhancers. However, data remain limited for traditional Central European products such as grilling sausages (Špekáčky) and frankfurters (Wiener Würstel), which are among the most frequently consumed processed meats in this region. Given their cultural and dietary importance, providing evidence on the feasibility of salt reduction in these products is of technological and public health relevance.”

We kindly inform the reviewer that the revised version of the manuscript is attached, with all modifications highlighted in yellow for ease of reference.

Reviewer 4 Report

Comments and Suggestions for Authors

Processed meat products are among those with high salt (sodium chloride) content. Consumer votes for lower salt intake. To answer the Consumer requirement the meat Processor uses different strategies with different, adverse sometimes, results. 

Sodium chloride in meat processing plays five specific roles - prolong product shelf life, enhance product flavor, taste and structure, increase muscle protein functionality.

The Authors discuss the results of reformulated meat product quality and safety produced with varying sodium chloride concentrations (1.4%, 1.6%, 1.8%, and 2.0%).

The topic fits the journal scope. The study was designed correctly and  sound technically.

Some general remarks regarding data interpretation in the Conclusion:

  • - The Conclusion should be justified.
  • - The statement "moderate reduction of salt does not significantly compromise product hygienic quality or sensory acceptability" (line 495) sounds not very novel. In cited papers (ex., Ruusunen, M.; Puolanne, E. Reducing sodium intake from meat products. Meat Sci. 2005, 70 (3), 531–541) low salt (1.4% NaCl) content in sausages was declared to be acceptable. Please be more specific in expressing the novelty of the research.
  • - The impact of varying sodium chloride concentrations in a complex of physical, chemical, technological properties and microbiological safety was aimed. Please present the results in conclusions to be interesting for the readership of the journal.
  • Based on your research, what NaCl concentration can you recommend for Špekáčky and Wiener Würstel to follow the consumer expectations?

Other recommendations.

Line 90 - please add how long meat samples were stored before processing, were meat samples packed?

lines 101-105 present the meat product formulation per 100 kg of product. If we add up all ingredients we will get ( 38.5 kg beef + 17.5 kg pork + 27.0 kg pork back fat + 2.5 kg potato starch + 0.85 kg spice mixture) 86.35 kg in total. Plus ice water - 23.0 kg, and the total weight is 109,35 kg. Please explain it. If water is added in cutter over the formulation, then all ingredients in total should be 100 kg.

Line 146 - please explain the sensory attribute "matter".

line 186 - please describe the equipment used.

You say in line 269 "fat tends to mask saltiness perception", but "Products with higher fat 
content generally exhibit a stronger salty taste" (line 288). This aspect needs clearer presentation.

line 428. Different superscripts in the columns within one product type indicate statistically significant differences. It seems necessary for Table 4. Please check data in other tables.

line 486 please check the spelling "comminated"

I suggest add data on meat (processed) meat product consumption while mentioning high popularity of these foods in Central Europe (lines 83, 94, 484).

Author Response

We would like to thank Reviewer 4 for the detailed evaluation of our manuscript and for providing constructive suggestions. These comments helped us to improve the clarity of the Conclusion, refine the methodological description, and strengthen the discussion of both sensory and technological aspects. In the revised version, we have addressed all points raised, including the clarification of ingredient formulation, sensory terminology, equipment specifications, and the role of fat in salt perception. We also emphasized the originality of our study, provided practical recommendations for NaCl concentrations in Špekáčky and Wiener Würstel, and supplemented the Introduction with additional data on processed meat consumption in Central Europe. Detailed responses to each comment are presented below.

Authors’ Response to Reviewer 4 Comments

Reviewer’s comment: The Conclusion should be justified and highlight novelty. Statement 'moderate reduction of salt does not significantly compromise product hygienic quality or sensory acceptability' is not novel. Please present complex results in conclusions and recommend NaCl concentration for Špekáčky and Wiener Würstel.

Thank you for this comment. We agree that the original conclusion was too general. In the revised manuscript, we have shortened the Conclusion and included specific results from sensory, physicochemical, and microbiological analyses. We emphasized the novelty of our study, which lies in its focus on traditional Central European products (Špekáčky and Wiener Würstel), for which detailed data on salt reduction have been lacking. We also added a practical recommendation: based on our results, a concentration of 1.6–1.8% NaCl can be considered optimal, balancing sensory acceptability, technological feasibility, and public health objectives.

Reviewer’s comment: Please add how long meat samples were stored before processing, were meat samples packed?

Thank you for this comment. We have added more detailed information about raw materials in the Methods section. We clarified that beef and pork were purchased from Jatka Ivančice, s.r.o. (CZ 242 ES, Czech Republic), transported in a refrigerated truck under chilled conditions, with the delivery temperature not exceeding +4 °C, stored at 2.5 °C, and processed within 48 h post-slaughter. The meat was neither packaged nor frozen prior to processing. These clarifications have been incorporated into the revised manuscript.

Reviewer’s comment: Ingredients sum to 109.35 kg instead of 100. Please clarify.

Authors’ Response / Odpověď autorů:

Thank you for this remark. The reported total weight (109.35 kg) exceeding 100 kg results from the fact that meat, fat, water, and starch are calculated per 100 kg of product (brutto), whereas spices and additives are traditionally expressed relative to 100 kg of mixture (g/kg). When recalculated into absolute weights, this leads to a higher overall sum. This is not an error but a standard technological practice. We would also like to emphasize that the recipe follows the legally defined specification for Špekáčky, which are recognized at the EU level as a Traditional Speciality Guaranteed (TSG) product (EU 2016). For this reason, the formulation was kept consistent with the officially protected recipe.

Reviewer’s comment: Please explain the sensory attribute 'matter'.

Thank you for this comment. In the revised manuscript, the attribute “matter” has been clarified and replaced with cohesiveness, which more accurately reflects the evaluated characteristic. This descriptor refers to the overall cohesiveness of the product mass when assessed on the cut surface and is part of the standard terminology used in Czech sensory evaluation practice.

Reviewer’s comment: Please describe the equipment used.

We thank the reviewer for this comment. In the revised manuscript, we have expanded the description of the analytical procedure and equipment used for sodium determination. Specifically, sodium content was measured by atomic absorption spectrometry using a contrAA 700 atomic absorption spectrometer (Analytik Jena, Germany) with air–acetylene flame atomisation. Prior to measurement, samples were digested in an Ethos SEL Microwave Labstation (Milestone, Italy) at 200 °C for 30 min after the addition of concentrated nitric acid and hydrogen peroxide. Data were processed with Aspect CS software, version 2.1, and Na-based salt content was calculated according to Regulation (EU) No. 1169/2011. We believe this additional information clarifies the methodology and fully addresses the reviewer’s request.

Reviewer’s comment: Clarify contradiction between 'fat masks saltiness' and 'higher fat = stronger salty taste'.

Thank you for this comment. We revised the text to clearly explain that the effect of fat on salt perception depends on the interaction between fat, protein, and salt in the product matrix. While fat generally reduces sodium release and thus attenuates perceived saltiness, products with a higher fat-to-protein ratio may appear relatively saltier, as protein has been shown to suppress saltiness perception. This mechanism has now been explicitly clarified in the Discussion, which resolves the initial apparent contradiction.

Reviewer’s comment: Different superscripts in the columns within one product type indicate statistically significant differences. 

Thank you for pointing this out. The table captions have been checked and revised to clearly state that different superscripts within one product type indicate statistically significant differences (P < 0.05).

Reviewer’s comment: Please check spelling 'comminated'.

Corrected – the proper spelling is 'comminuted'.

Reviewer’s comment: Please add data on processed meat consumption in Central Europe when discussing popularity.

Thank you for this suggestion. We have added current data on processed meat consumption in Central Europe to the Introduction. We clarified that the average intake of processed meat products in the region is 25–35 kg per capita annually, with sausages and frankfurters representing a major category and an important source of dietary sodium. These statements have been supported with new references. (In Central Europe, the average consumption of processed meat products remains high, ranging between 25 and 35 kg per capita annually, with sausages and frankfurters representing a dominant category. Reducing the salt content in these traditional products is challenging; nevertheless, they are often perceived positively from both a social and nutritional perspective (Waraczewski et al., 2023). In addition, consumer attitudes play a role in shaping dietary patterns. Respondents with stronger environmental concerns were more likely to adopt vegetarian or vegan diets, whereas more traditional and conservative consumers tended to prefer meat-based or flexitarian eating habits (Peri et al., 2025). Consequently, these products constitute a major dietary source of sodium for the population and represent a suitable model system for the evaluation of salt reduction strategies.

We kindly inform the reviewer that the revised version of the manuscript is attached, with all modifications highlighted in green for ease of reference.

Reviewer 5 Report

Comments and Suggestions for Authors

The study is interesting trying to address a major issue that the meat industry is facing.

Introduction

It is not focused. Information on cooked meat products should be added. Even information on the market trends could be useful. Infromation on clean label products is not relevant. 

Materials and methods

Materials and methods should be presented in the following order: product preparation, chemical composition, microbial analysis, colour and finally the sensory analysis.

All products have a low sodium chloride content. What is the typical salt content of commercial products? Is there a treatment that can be characterised as control?

Results and Discussion

Please present the finding in the same order as in materials and method since the reader should first read about differences in chemical composition, microbiology, instrumental colour measurements to associate them with the sensory analysis.

Colour analysis: Please consider adding Chroma (colour saturation) in your results.

Sensory analysis: What is matter? Can you please consider changing it with another word or provide an explanation?

Please present your results in a spider graph.

Please present your triangle analysis samples in a table or figure.

Conclusions: It is rather long. Make it more focused and refer to other analyses such as texture that should be conducted to have a comprehensive evaluation of the products.

Author Response

We would like to thank Reviewer 5 for the thorough review of our manuscript and for providing several constructive comments and suggestions. These remarks were very helpful in improving the focus of the Introduction, clarifying methodological details, and refining the presentation of the Results and Discussion. In the revised version, we have removed irrelevant content, added information on market trends and processed meat consumption, clarified the role of salt in cooked meat products, refined the terminology used in sensory evaluation, and adjusted the Conclusion to be more concise and practically oriented. We believe that these changes have improved the overall clarity, focus, and scientific value of the manuscript. Detailed responses to each comment are provided below.

Responses to Reviewer 5

Introduction

Reviewer’s comment: The introduction is not focused. Add information on cooked meat products and market trends. Clean label is not relevant.

Thank you for this constructive suggestion. In the revised version of the Introduction, we have restructured the text to ensure a clearer focus on salt reduction in cooked meat products, with specific emphasis on traditional Central European sausages such as frankfurters and grilling sausages. We removed the earlier section on “clean label” strategies, which was not directly relevant to the study’s objectives. Instead, we expanded the background on the functional role of salt in cooked meat products, its importance for flavour, preservation, and texture, as well as current knowledge regarding feasible reductions in cooked sausages and hams.

Furthermore, we incorporated up-to-date information on market relevance and consumption trends in Central Europe, highlighting that sausages and frankfurters remain dominant categories within processed meats, with per capita consumption between 25–35 kg annually. These additions align the Introduction more closely with the focus of the study and better justify why cooked sausages are an appropriate model system for evaluating salt reduction strategies.

We believe the revised Introduction now directly addresses the reviewer’s concerns, providing both technological context and market relevance while maintaining a clear focus on the research aim.

Materials and Methods

Reviewer’s comment: Materials and methods should be presented in the following order: product preparation, chemical composition, microbial analysis, colour and finally the sensory analysis.

We understand the reviewer’s suggestion regarding the preferred order of the Materials and Methods section. However, in this manuscript we chose to present the sensory evaluation at the end of the Methods because of its central importance and the relatively high number of panellists involved compared to similar studies. This structure allows the reader to first become familiar with the product preparation and analytical procedures, and then to follow the more extensive sensory methodology, which provides a key context for interpreting the subsequent results. We believe that this organisation improves the overall clarity and flow of the manuscript.

Reviewer’s comment: All products have low salt. What is the typical salt content of commercial products? Is there a treatment that can be considered control?

Thank you for this important question. In our study, the 2.0% NaCl concentration was chosen as the reference level, since it corresponds to a commonly applied salt content in commercial processed meat products, particularly frankfurters and grilling sausages (Desmond, 2006; Ruusunen & Puolanne, 2005). This selection was further supported by comparative data from Czech and German markets, which reported similar sodium levels in these product categories (Kameník et al., 2017). For this reason, the 2.0% formulation was considered the control treatment, allowing us to align the experiment with typical industrial practice and to evaluate the effects of stepwise salt reduction in relation to this standard benchmark.

Results and Discussion

Reviewer’s comment: Please present the findings in the same order as in materials and method since the reader should first read about differences in chemical composition, microbiology, instrumental colour measurements to associate them with the sensory analysis.

We appreciate the reviewer’s suggestion regarding the preferred sequence of the Results and Discussion. In this manuscript, however, we decided to present the sensory analysis at the end of the section, mirroring its placement in the Methods. This is because the sensory evaluation involved a relatively high number of panellists compared to similar studies, and it represents the most comprehensive part of our analysis. By structuring the Results and Discussion in this way, we allow the analytical findings (chemical, microbiological, colour) to provide context, after which the extensive sensory results can be more clearly interpreted. We believe this organisation improves the readability and logical flow of the manuscript.

Reviewer’s comment: Colour analysis: Please consider adding Chroma (colour saturation) in your results.

We thank the reviewer for this valuable suggestion. In our study, we focused on L*, a*, b* values and the total colour difference (ΔEab), which together provide a comprehensive description of colour changes in meat products. Since ΔEab integrates variation across all three parameters, the additional calculation of Chroma (C*) was considered redundant and unlikely to provide further interpretative value. Moreover, as the reviewer notes, C* is most informative when presented in combination with hue angle (h°), which was beyond the scope of this study. For these reasons, we retained the original set of colour parameters, which we believe sufficiently address the objectives of the work while maintaining clarity and focus in the results.

Reviewer’s comment: Sensory analysis: What is matter? Can you please consider changing it with another word or provide an explanation?

Thank you for this remark. The attribute “matter” has been clarified in the revised manuscript and replaced with “cohesiveness”, which more accurately describes the evaluated characteristic. This descriptor refers to the overall cohesiveness of the product mass when assessed on the cut surface

Reviewer’s comment: Please present your results in a spider graph.

Conclusion

Reviewer’s comment: Conclusions: It is rather long. Make it more focused and refer to other analyses such as texture that should be conducted to have a comprehensive evaluation of the products.

Thank you for this valuable comment. The Conclusion has been revised to be shorter and more focused on the main findings. We emphasized the novelty of the study, provided a clear recommendation for the optimal salt concentration (1.6–1.8% NaCl), and highlighted the technological and public health relevance of the results. In addition, we added a statement that future studies should include instrumental texture analysis to provide a more comprehensive evaluation of quality changes associated with sodium reduction.

We kindly inform the reviewer that the revised version of the manuscript is attached, with all modifications highlighted in grey for ease of reference.

Round 2

Reviewer 3 Report

Comments and Suggestions for Authors

The authors made changes based on my comments, so I recommend publishing the manuscript in its current form.